# Comparison of vaginal microbiota between women with inflammatory bowel disease and healthy controls

Ofri Bar[1,2], Leanna S. Sudhof[3,4], Laura J. Yockey[5], Agnes Bergerat[1], Nadav Moriel[2], Elizabeth Andrews[6], Ashwin N. Ananthakrishnan[3,6], Ramnik J. Xavier[3,7,8], Moran Yassour[9⊛], Caroline M. Mitchell[1,3,10⊛] *

1 Vincent Center for Reproductive Biology, Massachusetts General Hospital, Boston, MA, United States of America, 2 Department of Microbiology and Molecular Genetics, Faculty of Medicine, Hebrew University of Jerusalem, Jerusalem, Israel, 3 Harvard Medical School, Boston, MA, United States of America, 4 Department of Obstetrics & Gynecology, Beth Israel Deaconess Medical Center, Boston, MA, United States of America, 5 Departments of Medicine, Massachusetts General Hospital, Boston, MA, United States of America, 6 Departments of Gastroenterology, Massachusetts General Hospital, Boston, MA, United States of America, 7 Departments of Molecular Biology and Center for Computational and Integrative Biology, Massachusetts General Hospital, Boston, MA, United States of America, 8 Broad Institute, Boston MA, United States of America, 9 The Rachel and Selim Benin School of Computer Science and Engineering, Hebrew University of Jerusalem, Jerusalem, Israel, 10 Departments of Obstetrics & Gynecology, Massachusetts General Hospital, Boston, MA, United States of America

⊛ These authors contributed equally to this work.
* Caroline.mitchell@mgh.harvard.edu

**Data Availability Statement:** The sequences for this project have been uploaded to the Short Read Archive under BioProject Accession PRJNA849603.

## Abstract

### Background

The gut microbiota in patients with inflammatory bowel disease are perturbed in both composition and function. The vaginal microbiome and its role in the reproductive health of women with inflammatory bowel disease is less well described.

### Objective

We aim to compare the vaginal microbiota of women with inflammatory bowel disease to healthy controls.

### Methods

Women with inflammatory bowel disease enrolled in a longitudinal cohort study provided self-collected vaginal swabs. Healthy controls underwent provider-collected vaginal swabs at routine gynecologic exams. All participants completed surveys on health history, vulvovaginal symptoms and gastrointestinal symptoms, if applicable. Microbiota were characterized by sequencing the V4 region of the 16S rRNA gene. Associations between patient characteristics and microbial community composition were evaluated by PERMANOVA and Principal Components Analysis. *Lactobacillus* dominance of the microbial community was compared between groups using chi-square and Poisson regression.

**Funding:** This work was supported by the the Domolky Innovation Award (CM), the Division of Intramural Research, National Institute of Allergy and Infectious Diseases R21AI113439 (CM), the National Institute of Diabetes and Digestive and Kidney Diseases DK043351 (RX), Center for Microbiome Informatics and Therapeutics Flagship Project (RX).

**Competing interests:** Dr. Mitchell has received research funding from Scynexis, Inc. Dr. Xavier is co-founder of Jnana Therapeutics and Celsius Therapeutics, and is a consultant to Nestle. The remaining authors report no conflict of interest. This does not alter our adherence to PLOS ONE policies on sharing data and materials.

## Results

The cohort included 54 women with inflammatory bowel disease (25 Ulcerative colitis, 25 Crohn's Disease) and 26 controls. A majority, 72 (90%) were White; 17 (31%) with inflammatory bowel disease and 7 (27%) controls were postmenopausal. The composition of the vaginal microbiota did not vary significantly by diagnosis or severity of inflammatory bowel disease but did vary by menopausal status (p = 0.042). There were no significant differences in Shannon Diversity Index between healthy controls and women with IBD in premenopausal participants. There was no difference in proportion of *Lactobacillus* dominance according to diagnosis in premenopausal participants. A subgroup of postmenopausal women with Ulcerative colitis showed a significant higher alpha diversity and a lack of *Lactobacillus* dominance in the vaginal microbiome.

## Conclusions

Menopausal status had a larger impact on vaginal microbial communities than inflammatory bowel disease diagnosis or severity.

## Introduction

Inflammatory bowel disease (IBD) affects about 600 per 100,000 people in North America, and worldwide the incidence is increasing [1]. The etiology of this group of chronic inflammatory conditions is not completely understood, and disease manifestations are likely due to a complex interplay of underlying host genetic factors, environmental perturbations, and the gut microbiota [2, 3]. Gene loci associated with risk of IBD are part of pathways important in epithelial barrier function, the innate immune response, autophagy, and antimicrobial activity, among others. The gut microbiota in patients with IBD, though heterogeneous, has some consistent aberrant features, such as higher abundance of *Enterobacter* species and lower abundance of *Clostridia* and *Bacteroides* species [2].

As the understanding of the vital role that gut microbiota play in health and disease states has matured, the recognition of the role that vaginal microbiota may play in reproductive health has also grown [4–6]. IBD is associated with well-described extra-intestinal complications, including at environmental interfaces of the skin and oral mucosa. We report an increase in prevalence of vulvovaginal symptoms in IBD, but the mechanism of these symptoms is not described [7]. We hypothesized that patient characteristics leading to an impaired host-microbe relationship in the gut in patients with IBD might influence the microbiota in other mucosae, such as the vaginal mucosa. Compared to bacterial communities at other body sites, the diversity of microbial species in the vagina is limited; in many healthy women *Lactobacillus* species make up over 90% of the microbial community [8–10]. *Lactobacillus* dominance is associated with favorable reproductive outcomes, including decreased rates of prematurity, vaginitis, genital inflammation, and risk of HIV acquisition [11–16].

The vaginal microbiota and vulvovaginal health of women with IBD have not been well studied. Two studies have shown concurrence of vaginal complaints with intestinal symptoms in women with IBD [7, 17]. A prospective study of 98 women in Europe showed higher detection of *Gardnerella vaginalis* biofilm on epithelial cells in urine specimens from women with IBD compared to healthy controls but did not find an association between *G. vaginalis* and IBD disease activity or concurrent medication use [18]. In this pilot study, we aimed to

characterize the vaginal microbial communities in patients with IBD compared to healthy controls, and to assess associations between vaginal microbiome and severity of IBD.

## Materials and methods

This pilot, case-control study included participants with IBD enrolled between March 2015 and February 2018 in a longitudinal cohort study at Massachusetts General Hospital who were classified as having Crohn's disease (CD), ulcerative colitis (UC) or indeterminate colitis (IC) and opted in to the vaginal microbiota sub-study [19]. Samples collected from healthy controls at a routine gynecologic exam between October 2014 and September 2016 at the same institution were identified from an existing repository. As both race and menopausal status have been associated with differences in vaginal microbiota [9, 20], control participants were matched to case samples by age, race and menopausal status when possible, though there were insufficient numbers of samples to match exactly. Use of samples from each cohort was approved by the Partners Institutional Review Board (IRB # 2004P001067; 2014P001066).

Study participants with IBD performed self-collected vaginal swabs at home and sent them to the lab in a dry tube by mail. Participants collected between 1–3 swabs, at intervals ranging from 9–264 days apart (median 106) depending on the duration of their participation in the parent study and number of follow-up visits while this vaginal microbiota sub-study was ongoing. Samples collected and processed in this way have been shown to have similar results as swabs collected by clinicians and transported directly to the lab [21]. Samples from healthy controls were provider-collected vaginal swabs from a single time point at a routine gynecological exam.

Written surveys on health history, current medications, vulvovaginal symptoms, and IBD symptoms, if applicable, were completed by all participants (cases and controls). Menopause was determined according to the participants report. The Harvey Bradshaw Index was used to classify Crohn's disease activity (< 5 Remission, 5–16 Active, > 16 Severe), and the Simple Colitis Activity Index for Ulcerative and Indeterminate colitis (< 5 Remission, ≥ 5 Active) [22, 23]. Questionnaires about vulvar and vaginal symptoms asked about itching, burning, irritation, dryness, pain and discharge [24]. For participants who provided more than one swab, we calculated a score for the change in disease severity between visits (later visit score—earlier visit score) to identify intervals with an improvement (negative score) or worsening of disease (positive score).

The swabs were frozen at -80˚C. Elution of the sample and DNA extraction were performed as previously described [25]. Sequencing of the V4 region of the 16S rRNA gene was performed using Illumina MiSeq and ASV were assigned using DADA2 [26]. The Assign Taxonomy function was used to implement RDP Naive Bayesian Classifier algorithm as previously described [27] with kmer size 8 and 100 bootstrap replicates.

Abundant sequences that were not assigned a taxonomy to the species level were put into BLAST [28] for identification, using 99% homology as a threshold to assign identity (sequences and BLAST results in S1 Table). If sequence identity could not be resolved by this method the ASV was considered unclassified. There were an average of 43,978 reads per sample (range 8025–168555). Taxa that were present in less than 5% of the samples was removed.

We compared demographic and clinical characteristics between women with vs. without IBD using chi square or Student's t-test as appropriate. Because people with IBD may have provided more than one sample, while healthy controls only provided a single sample we conducted 2 analyses: a cross sectional comparison of one randomly selected sample from cases to one sample from controls and a longitudinal analyses within cases (using all samples) to see if there were changes in microbiota related to underlying changes in disease symptoms.For the

cross-sectional analyses, when an IBD cases had more than one sample, a random sample was selected using a predetermined seed to allow reproducibility of the results. Associations between patient characteristics and microbiota were assessed using permutational multivariate analysis of variance ANOVA (PERMANOVA) [29].

The R software was used for statistical analysis, using Ape and Vegan packages. Alpha diversity was assessed using the Shannon Diversity Index and was calculated using Diversity() function which is part of the Vegan package. Beta diversity was calculated using the Vegdist() and Pcoa() functions. Principal Coordinates Analysis (PCoA) was used to examine similarities or dissimilarities in microbial composition by diagnosis, menopausal status and dominant *Lactobacillus* species.

Since the diversity of the vaginal microbiome is limited and *Lactobacillus* species make up over 90% of the microbial composition, vaginal microbial communities generally cluster according to the dominant *Lactobacillus* species. Some studies cluster samples by Community State Type (CST) [9], and others use a *Lactobacillus* relative abundance threshold to classify communities [11, 13]. We assigned a categorical variable based on relative abundance of *Lactobacillus* species in the following manner: if the *Lactobacillus* species was the most abundant species in the sample, and also had at least 40% relative abundance, it was defined as the dominant species. Because *L. iners* is not considered as beneficial as other vaginal *Lactobacillus* species, for some analyses we also classified samples as either *L. iners* dominant or non-*iners* *Lactobacillus* dominant. If *L.iners* had at least 50% relative abundance the community was classified as *L. iners* dominant. Using all samples from premenopausal participants, overall *Lactobacillus* dominance (as a categorical variable) was compared across IBD status using a mixed effects logistic regression, with participant ID as a grouping variable to account for multiple samples from some participants.

## Results

A total of 54 participants with IBD and 26 healthy controls were enrolled in the study. Of the participants with IBD, 26 provided more than one swab (10 provided 2, 16 provided 3), thus the study included a total of 122 swabs. For cross-sectional analyses, when a single random swab was chosen from each of the 54 participants with IBD, the sample size was reduced to a total of 80 samples.

The participants were predominantly White and ranged in age between 30 and 53 (Table 1). Approximately a quarter of participants in each group were postmenopausal. The majority were diagnosed with either Crohn's Disease (46%) or Ulcerative Colitis (47%). At enrollment, half of IBD patients (56%) were in remission. Women with IBD were more likely to report moderate-severe vulvovaginal symptoms: 17/51 (33%) vs. 3/26 (12%) (p = 0.04). There were no significant differences in vulvovaginal symptoms according to IBD disease activity across all visits, though there was a trend to lower prevalence when people had active disease: remission 32% (29/73) vs. active disease 17% (2/12), (p = 0.30). Only 3/54 (6%) people with IBD reported a history of perianal disease or vulvar fissures.

We conducted a PERMANOVA analysis using a single sample for each participant (i.e. a randomly selected swab for the IBD case group) to assess the relative influence of menopause or IBD diagnosis on vaginal microbial community composition. Menopausal status (F = 2.06, p = 0.042) had a much greater effect than diagnosis of IBD (F = 1.22, p = 0.192), and so most subsequent analyses were conducted stratified by menopausal status.

The majority of premenopausal participants had low-diversity, *Lactobacillus*-dominant microbial communities (Fig 1A). Postmenopausal women were less likely to have *Lactobacillus*-dominant communities (Fig 1B). Principal coordinate analysis (PCoA) of Bray–Curtis

**Table 1. Demographic and symptomatic characteristics of study participants at the time of first vaginal swab collection.**

| Characteristic | | Women with IBD (N = 54) | Women without IBD (N = 26) |
|---|---|---|---|
| Median age in years (IQR) | | 40 (30, 52) | 42 (31, 48) |
| Race | White | 48 (89%) | 24 (92%) |
| | Black | 2 (4%) | 1 (4%) |
| | Asian | 1 (2%) | 1 (4%) |
| | Other | 2 (7%) | — |
| Hormonal birth control use | | 14 (26%) | 2 (8%) |
| Menopause | | 14 (26%) | 7 (27%) |
| Moderate-severe vulvovaginal symptoms | | 18 (33%) | 3 (12%) |
| IBD diagnosis | Crohn's disease | 24 (44%) | - |
| | Ulcerative colitis | 25 (47%) | - |
| | Indeterminate colitis | 5 (9%) | - |
| Current IBD therapy* | 5-ASA derivative | 8/21 (38%) | - |
| | Steroid | 5/21 (24%) | - |
| | Antibiotics | 2/21 (10%) | - |
| | Probiotic | 1/21 (5%) | - |
| | Immunosuppressant | 8/21 (38%) | - |
| | Biologic | 7/21 (33%) | - |
| IBD activity by HBI or SCAI | In remission | 46 (85%) | - |
| | Active | 8 (15%) | - |

IBD: Inflammatory bowel disease; 5-ASA 5-acetylsalycilic acid;HBI Harvey-Bradshaw Index; SCAI Simple Colitis Activity Index

*Incomplete data available

dissimilarities demonstrated no association between diagnosis and vaginal microbial community composition (Fig 2A). Three main clusters were seen: non-*Lactobacillus* dominant, and two *Lactobacillus*-dominant clusters which were defined primarily by the species most dominant in those samples, either *L. crispatus* or *L. iners* (Fig 2B).

We then compared alpha diversity by IBD diagnosis, stratified by menopausal status, using the Shannon Diversity Index (SDI) (Fig 3). There were no differences between healthy controls and women with IBD in premenopausal participants. Among postmenopausal participants, those with Ulcerative Colitis had a significantly higher SDI compared to healthy controls (p = 0.014), but there was not a significant difference for those with Crohn's Disease. We also compared SDI across IBD symptom severity for all visits and did not see a significant difference (S1 Fig).

We next sought to determine the association between IBD diagnosis and *Lactobacillus* dominance. We compared samples according to the *Lactobacillus* categories described above (Fig 4). There was no difference in proportion of *Lactobacillus* dominance according to diagnosis in premenopausal participants (Table 2). All samples from postmenopausal participants with Ulcerative Colitis demonstrated a non- *Lactobacillus* dominant vaginal microbiota and all postmenopausal healthy controls had *Lactobacillus* dominant communities, which was significantly different (p < 0.01). There were no significant differences in *Lactobacillus* status between postmenopausal controls and women with Crohn's Disease.

Of the participants with IBD, 26 provided more than one swab (10 provided 2, 16 provided 3). We examined if changes in IBD severity over time were associated with changes in the diversity of the vaginal microbiome in each woman. Using change in IBD severity score between visits, participants were categorized as no change, improvement or deterioration. We then calculated the change in SDI between those vaginal samples. We did not observe any

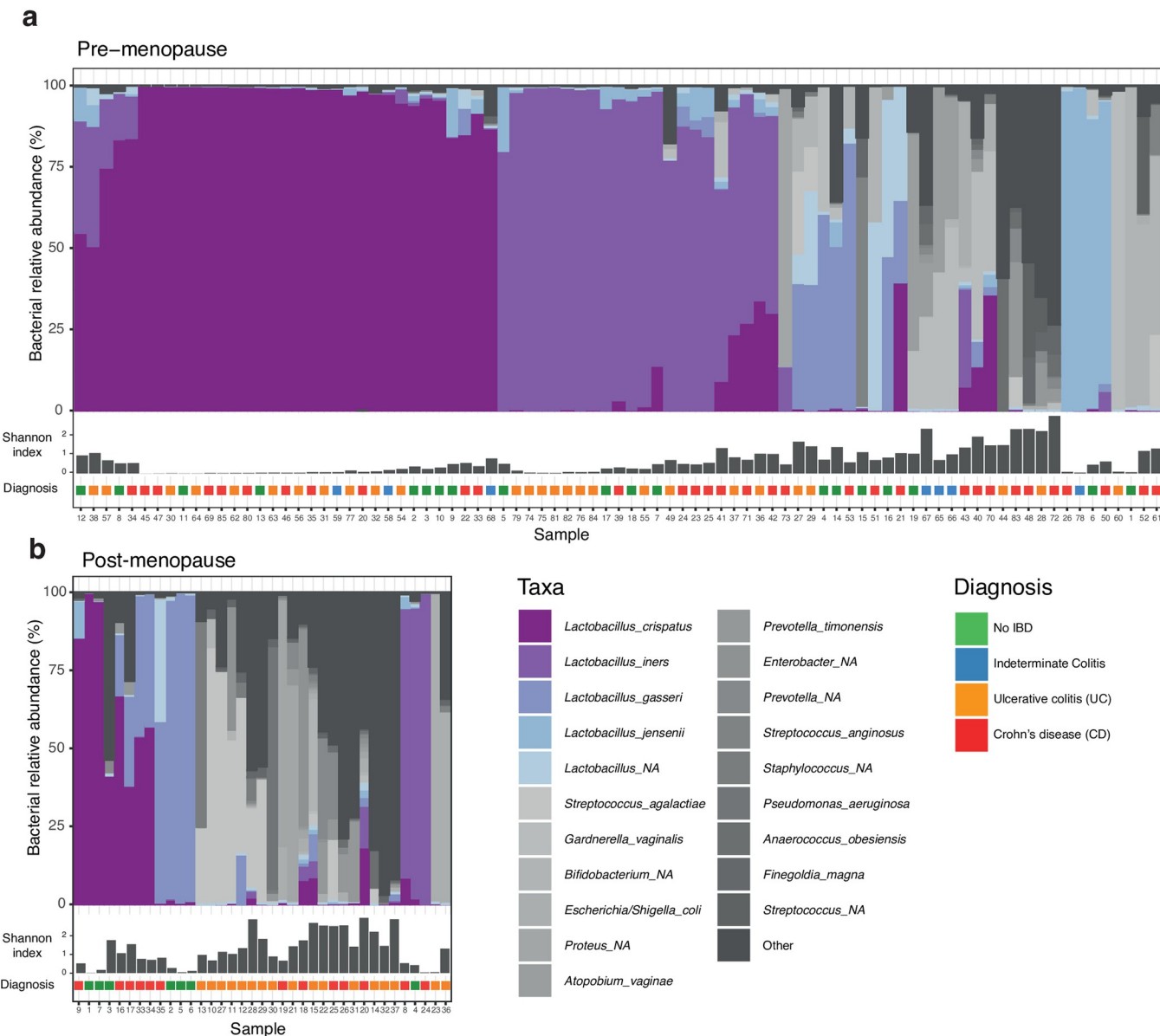

**Fig 1. Composition of vaginal microbiota by menopausal status and IBD diagnosis.** Stacked bar plots indicating the bacterial relative abundance of the 20 most prevalent taxa in the vaginal microbiota in 122 samples from 80 women. Samples were stratified according to menopausal status. The order of samples in the stacked bar plot is determined by hierarchical clustering (see heatmap in S1 Fig). Inflammatory bowel disease diagnosis status is shown above each sample.

difference in change in SDI between intervals with improvement or deterioration in the severity index score (Fig 5).

## Discussion

In this pilot study of both pre- and postmenopausal participants we found limited evidence that vaginal microbiota are different in participants with IBD compared to healthy controls. Among our postmenopausal participants, people with Ulcerative Colitis were much less likely to have *Lactobacillus* dominance, and more likely to have a higher diversity of the vaginal microbial community. However, no differences were seen between women with vs. without

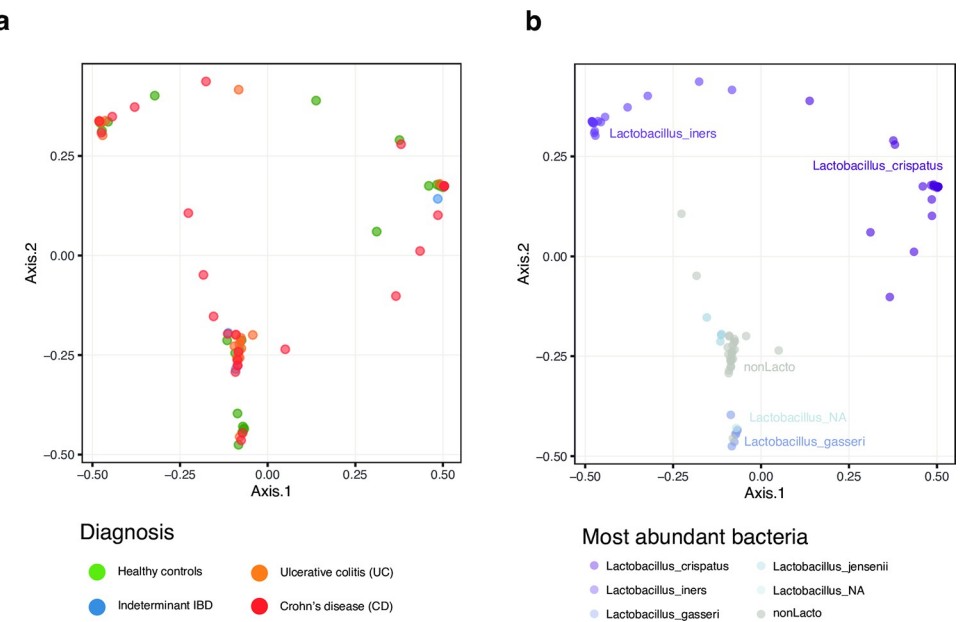

**Fig 2. Comparison of community composition.** Principal Coordinates Analysis was used to compare beta-diversity of samples. Figures are colored by (a) inflammatory bowel disease diagnosis, and b) dominant species. For this analysis a single swab was randomly selected for each participant (n = 80).

IBD among premenopausal participants. We did not find any difference in the report of moderate-severe vulvovaginal symptoms between participants with vs. without IBD, and did not see an association between vaginal microbial diversity and IBD symptom severity.

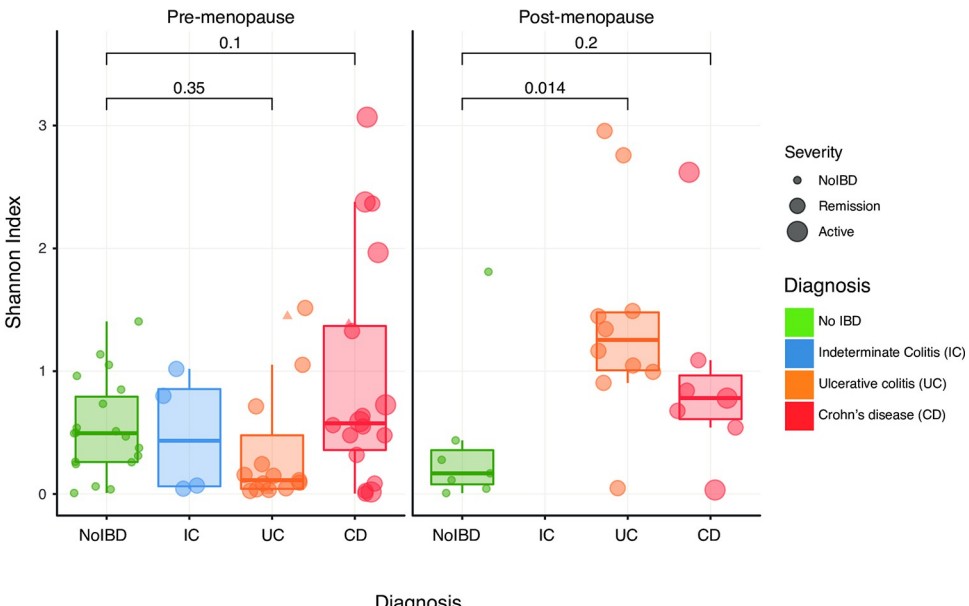

**Fig 3. Comparison of community diversity.** Shannon Diversity Index was used to compare alpha diversity by inflammatory bowel disease diagnosis, stratified by menopausal status. Gastrointestinal symptom severity was assigned using either the Harvey Bradshaw Index (Crohn's) or Simple Clinical Colitis Assessment (Ulcerative Colitis). For this analysis a single swab was randomly selected for each participant (n = 80).

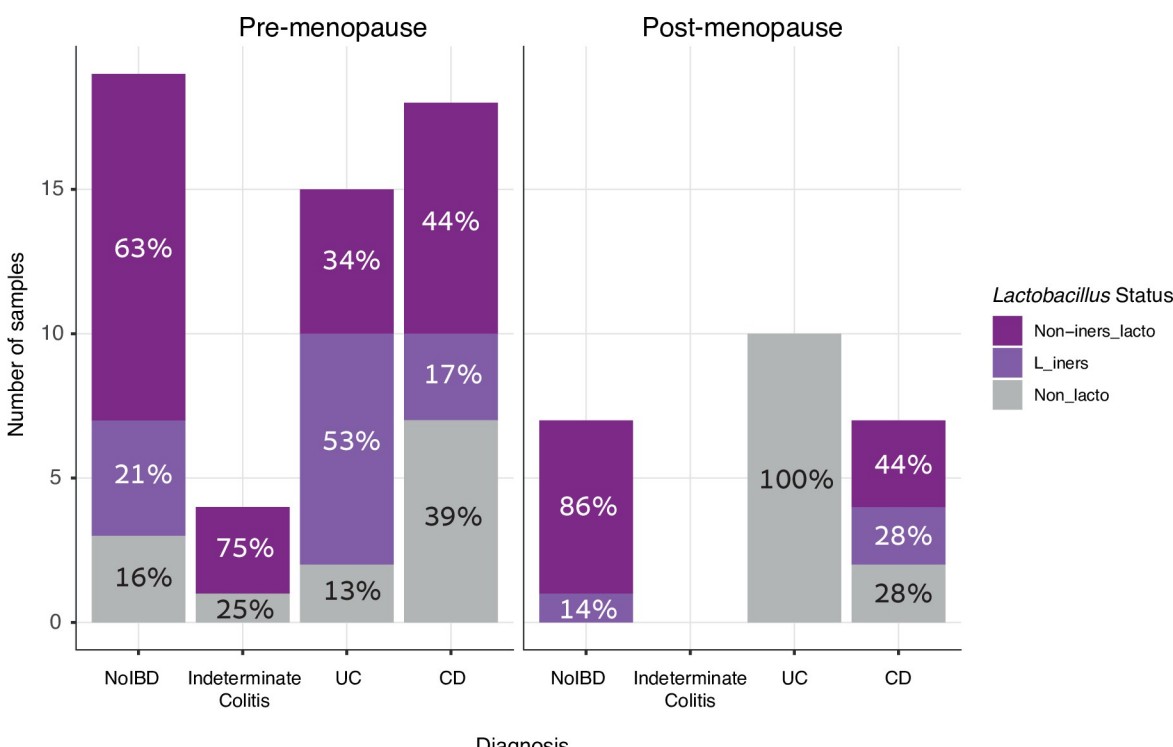

**Fig 4. Comparison of *Lactobacillus* dominance.** The number and proportion of samples dominated by *Lactobacillus iners*, non-*iners Lactobacillus* or non-*Lactobacillus* species was compared by inflammatory bowel disease diagnosis, stratified by menopause. A sample was considered *Lactobacillus* dominant if > 40% of sequences were from *Lactobacillus*, and considered *L. iners* dominant if > 50% of all *Lactobacillus* sequences were from *L. iners*. For this analysis a single random sample was selected for each participant (n = 80).

Our results contrast with the previously described association between IBD diagnosis and increased urinary detection of *Gardnerella*, a vaginal bacteria associated with high-diversity vaginal communities, in urine [18]. A few studies examining changes in vaginal microbiota over time in pregnant participants with IBD have not found significant differences from what has been described in pregnant people without IBD, aside from higher prevalence of *Mollicutes* in one cohort [30, 31]. In general, women with IBD have similar reproductive outcomes as women without IBD, especially when disease is quiescent. Active disease in pregnancy is associated with higher rates of complications such as low birth weight, preterm birth and miscarriage (reviewed in [32]). In our study we did not see an association between disease severity

**Table 2. Association between inflammatory bowel disease diagnosis or severity and *Lactobacillus* dominance, across all swabs, in premenopausal women.**

| Sample Characteristic | | *Lactobacillus* dominant | OR* |
|---|---|---|---|
| Diagnosis | No IBD | 16/19 (84%) | Reference |
| | CD | 22/34 (65%) | 0.05 (0.001, 2.56) |
| | UC | 22/29 (76%) | 0.24 (0.01, 8.74) |
| | IC | 4/7 (57%) | 0.17 (0.001, 30.08) |
| Disease severity | Remission | 38/54 (70%) | Reference |
| | Active | 5/9 (56%) | 0.72 (0.02, 22.54) |

IBD Inflammatory Bowel Disease; CD Crohn's Disease; UC Ulcerative colitis; IC Indeterminate colitis; IRR Incidence Rate Ratio

*Mixed-effects logistic regression to account for multiple samples from some participants

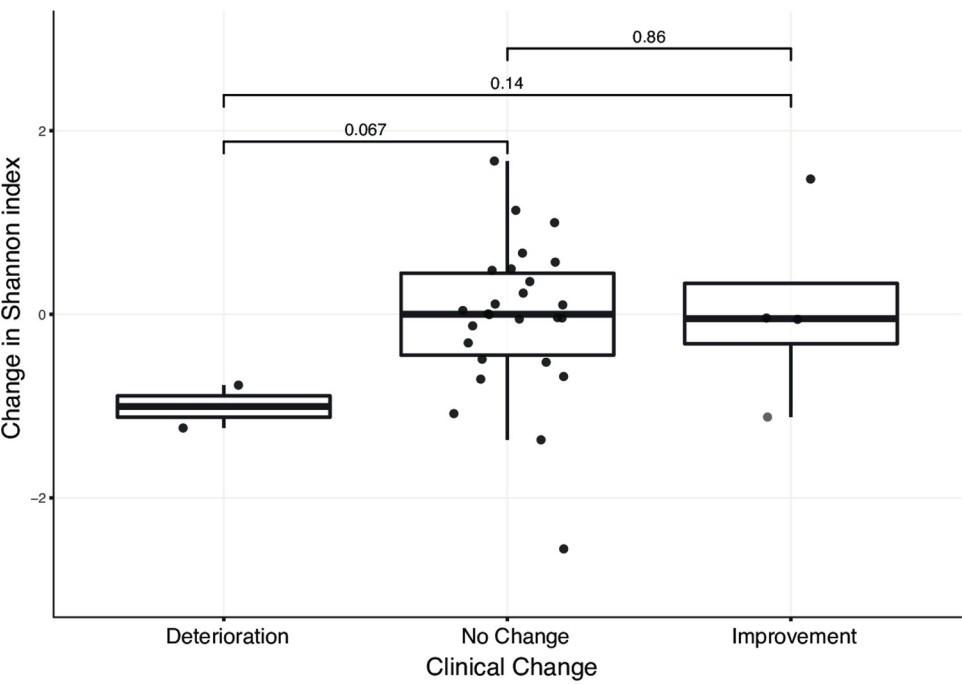

**Fig 5. Change in community diversity and gastrointestinal symptom severity.** For the 26 participants with inflammatory bowel disease who contributed more than one swab, we compared symptom severity between visits and assigned each interval as "improved", "stable" or "worsened." The change in Shannon Diversity Index across the same interval was compared between the three categories.

and vaginal microbial diversity, either cross-sectionally or longitudinally. It is possible that our study was too small to see subtle differences between groups. There are no other studies exploring differences in vaginal microbiota between women with vs. without IBD.

Many studies have reported that the composition of the gastrointestinal microbiota in IBD is altered compared to that of healthy subjects, and has greater variability over time [33–35]. At least one study has shown that alterations in gut microbiota are associated with severity of disease [36]. Furthermore, in almost all of the experimental models for IBD, under germ-free conditions disease either does not develop at all or is significantly attenuated, suggesting that microbes are essential for the development of intestinal inflammation in IBD [37]. The most consistent changes seen in people with IBD are a reduction in the diversity of gut microbiota and the lower abundance of Firmicutes [38, 39]. Furthermore, it has been discovered that some genes and gene loci that are associated with IBD, encode important components for sensing and adapting to changes in the gut microbiome [40]. Aside from impaired microbial sensing ability and bacterial clearance, IBD gene variants may also alter intestinal immune homeostasis by disrupting the intestinal epithelial barrier [34]. In contrast to the gut, where a diverse microbiome is considered beneficial, a healthy vaginal microbiome has low diversity and is dominated by lactobacilli.

In the vaginal microbiota of participants in our study, there does not appear to be a reduction in prevalence of lactobacilli, which are the most common vaginal Firmicute. The exception to this was in the subset of postmenopausal women with Ulcerative colitis. However, with such small numbers it is challenging to ascertain whether the difference in microbiota was due to menopausal status or underlying IBD. More of our participants with IBD reported moderate-severe vulvovaginal symptoms, which may indicate greater mucosal inflammation in the presence of a similar microbial community. We were unable to assess

differences in soluble vaginal fluid immune factors because samples were stored and shipped at room temperature.

Our data suggest that in people with IBD there is less impact on the vaginal microbiota than on the gut microbiota. Premenopausal women with IBD should be evaluated for vaginitis when symptoms are present, just as with any patient. The low prevalence of *Lactobacillus* seen in postmenopausal women with IBD could put them at higher risk for recurrent UTI.

While we did not see a difference in vaginal microbiota, women with IBD were more likely to report moderate to severe vaginal symptoms. We hypothesize that a difference might be found in the vaginal mucosal immune function rather than in the vaginal microbiome and so a potential next step could be to perform a study with a larger cohort that compares both the vaginal microbiota and the vaginal mucosal immune function in patients with IBD to patients without IBD. It may also be important to correlate gut vs. vaginal microbiota. If indeed there is a difference in the mucosal immune function then the vagina may provide an additional platform for gaining a better understanding of the pathophysiology of IBD as it is physically more accessible and has a more homogeneous microbiome than the gut. Thus, if there are perturbations in mucosal immune function that are universal in patients with IBD, these may be easier to identify in the vaginal mucosa.

## Strengths and weaknesses

Our study had a relatively small sample of patients, though large enough to detect a difference in vaginal microbiota between pre- and postmenopausal women. Menopausal status was an effect modifier and in the stratified analysis our numbers were likely too small to detect differences between groups. This suggests that differences between women with vs. without IBD are not as pronounced as those between pre- and postmenopausal women. We did not have sufficient stored samples to match equal numbers of healthy controls to cases, however the proportions of pre- vs. post-menopausal people were similar in the two groups. Because our participants with IBD mailed their swabs from home, we were not able to assess markers of immune response, which are far less stable than DNA. It is possible that people with IBD have a different mucosal immune response to the same microbiota, which could drive symptoms or adverse outcomes.

## Conclusion

Menopausal status had a larger impact on vaginal microbial communities than inflammatory bowel disease diagnosis or severity.

## Supporting information

**S1 Fig. Comparison of diversity according to severity.** Shannon Diversity Index was used to compare alpha diversity by symptom severity. Gastrointestinal symptom severity was assigned using either the Harvey Bradshaw Index (Crohn's) or Simple Clinical Colitis Assessment (Ulcerative Colitis). For this analysis a single random sample was selected for each participant (n = 80).
(TIF)

**S1 Table. Sequences assigned taxonomy by BLAST.**
(XLSX)

## Acknowledgments

The findings were presented in part at the IDSOG 2018 Annual Meeting in Philadelphia, PA, on 8/2-8/4/2018.

The sequences for this project have been uploaded to the Short Read Archive under BioProject Accession PRJNA849603.

## Author Contributions

**Conceptualization:** Ashwin N. Ananthakrishnan, Ramnik J. Xavier, Caroline M. Mitchell.

**Data curation:** Leanna S. Sudhof, Laura J. Yockey, Moran Yassour.

**Formal analysis:** Ofri Bar, Leanna S. Sudhof, Laura J. Yockey, Nadav Moriel, Moran Yassour.

**Funding acquisition:** Ramnik J. Xavier, Caroline M. Mitchell.

**Investigation:** Agnes Bergerat, Elizabeth Andrews, Ashwin N. Ananthakrishnan, Ramnik J. Xavier.

**Methodology:** Ramnik J. Xavier, Caroline M. Mitchell.

**Project administration:** Elizabeth Andrews, Caroline M. Mitchell.

**Supervision:** Agnes Bergerat, Ashwin N. Ananthakrishnan, Ramnik J. Xavier, Moran Yassour, Caroline M. Mitchell.

**Visualization:** Ofri Bar, Laura J. Yockey, Nadav Moriel, Moran Yassour.

**Writing – original draft:** Ofri Bar, Leanna S. Sudhof.

**Writing – review & editing:** Ofri Bar, Leanna S. Sudhof, Laura J. Yockey, Agnes Bergerat, Nadav Moriel, Elizabeth Andrews, Ashwin N. Ananthakrishnan, Ramnik J. Xavier, Moran Yassour, Caroline M. Mitchell.

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
