## [Decision Letter · Decision Letter 0]

13 Mar 2023

PONE-D-23-01900

Comparison of vaginal microbiota between women with inflammatory bowel disease and healthy controls

PLOS ONE

Dear Dr. Mitchell,

Thank you for submitting your manuscript to PLOS ONE. After careful consideration, we feel that it has merit but does not fully meet PLOS ONE’s publication criteria as it currently stands. Therefore, we invite you to submit a revised version of the manuscript that addresses the points raised during the review process.

We look forward to receiving your revised manuscript.

Kind regards,

José António Baptista Machado Soares, PhD

Academic Editor

PLOS ONE

Journal Requirements:

   "This work was supported by the Domolky Innovation Award (CM), NIAID R21AI113439-01A1 (CM), and DK043351, Center for Microbiome Informatics and Therapeutics flagship (RX). The funders had no input into the design, conduct or analysis of the study. "

   "I have read the journal's policy and the authors of this manuscript have the following competing interests: Dr. Mitchell has received research funding from Scynexis, Inc. Dr. Xavier is co-founder of Jnana Therapeutics and Celsius Therapeutics, and is a consultant to Nestle. The remaining authors report no conflict of interest."

Additional Editor Comments:

Dear authors,

I read the two reviewers' reports and I agree with them. The manuscript needs to address the following concerns:

The present version of the manuscript lacks important information on classification of vaginal microbiome composition described earlier and discussion is overstating the resultsAddress all concerns of Reviewer 1 in the specfic comments in his/her reportI believe that the present study is worth to be publish and it is important to evaluate the interaction and dynamics of gut and vaginal microbiota. However, major revision is needed before consideration for publication endorsement.

Reviewers' comments:

Reviewer's Responses to Questions

**Comments to the Author**

1. Is the manuscript technically sound, and do the data support the conclusions?

Reviewer #1: Yes

Reviewer #2: Partly

2. Has the statistical analysis been performed appropriately and rigorously? 

Reviewer #1: Yes

Reviewer #2: Yes

3. Have the authors made all data underlying the findings in their manuscript fully available?

Reviewer #1: Yes

Reviewer #2: No

4. Is the manuscript presented in an intelligible fashion and written in standard English?

Reviewer #1: Yes

Reviewer #2: Yes

5. Review Comments to the Author

Reviewer #1: Patients with IBD often complain about associated genital discomfort. The authors investigated whether this is related to a potential dysbiosis of the vaginal microbiome in relation to IBD status.

In the current study, 54 women with IBD and 26 healthy controls were included. Assessment of clinical parameters, including IBD status, and vaginal swabs with identification of their microbiota composition was performed. The microbiota composition was then correlated with clinical parameters.

The authors concluded that there was no apparent relationship between IBD status and the vaginal microbiome. However, the vaginal microbiome was strongly related to the menopausal status of the women.

In summary, the authors try to answer a relevant question for clinical practice. The manuscript is well written and outlines the research and conclusion in a concise manner. We recommend the manuscript can be published with a minor revision centered on providing additional details as specified below.

Minor issues:

- A subgroup of postmenopausal women showed a relationship between UC and the vaginal microbiome in regard to a significant difference in alpha diversity and lack of Lactobacillus dominance. These results are not mentioned in the abstract, which might hide a relevant difference in a UC subgroup. We however agree with the central statement that menopausal status had a larger impact on vaginal microbial communities than IBD status.

- Line 177: Instead of a Poisson model for microbiome studies often a negative binominal model is used because microbiome data are overdispersed. Also, the parameters of the model are not given. It would be good to explain the choice of the model and provide details of the model formula. Please state which parameters were used in the model formula (timepoint? IBD status? Patient ID?)

- Line 151: Please provide information given about which bacteria database was used for taxonomy matching to the ASVs before the BLAST step (Greengenes, SILVA, custom?)

- Line 211: Because vaginal swabs are less diverse than fecal samples, samples cluster mainly in regard to the major Lactobacillus subtype/ non-Lactobacillus status. This could also be mentioned in the text to clarify for the readers.

- Line 288: The discussion lacks any consideration of the health of the vaginal microbiome. In contrast to the gut microbiota where a higher diversity is generally regarded as beneficial, this might not be the case for the vaginal microbiome, as a diverse microbiota could be an indication of bacterial vaginosis. Thus, an increased alpha-diversity in a subgroup (potentially by cross-contamination) is a hypothetical pathogenetic mechanism in accordance with the presented results in postmenopausal UC patients. We propose to discuss this also in the text

Reviewer #2: The article “Comparison of vaginal microbiota between women with inflammatory bowel disease and healthy controls” describes bacterial composition of vaginal samples in diseased (n=50) and non-diseased women (n=26). Generally, the findings of the study are interesting; especially as they rather argue for no specific involvement of bacterial composition being involved in the more severe vulvovaginal symptoms reported by IBD patients. However, this study lacks important information on classification of vaginal microbiome composition described earlier and discussion is a bit overstating the overall findings. Thus, overall, this study has merit if specific concerns (please see below) can be addressed.

Methods

L151: The authors should indicate which classifier and database were used for ASV assignment as well as parameters set in the DADA2 pipeline applied.

LL168-177. The authors should indicate which software was used for analysis of alpha- and beta-diversity. In general, this section is too short to be able to reproduce the analysis of raw data and thus, should either be specified with parameters used (filtering of low abundant samples, evaluation of controls, etc.) or provide codes used in an appropriate online repository, e.g. Github.

Results

LL211-213: The existence of those clusters has been described earlier and have been classified as Community state types (CSTs) (https://www.ncbi.nlm.nih.gov/pmc/articles/PMC3063603/). This information should be added and probands should be classified in those types.

Discussion

LL321-323: Since this finding is based on a very low number of samples, the authors should avoid to give any therapy indications but rather stick to the follow-up studies with more individuals included as suggested in the following section.

LL329-333: This is very speculative – could the authors include some literature underlining this hypothesis? Particularly, as the authors could not detect any significant changes between active IBD and remission, this seems to be overstated.

LL335-345: pH is the most common factor that at least drive differences between Lacto and non-Lacto groups. Thus, it should be included within the next study.

6. PLOS authors have the option to publish the peer review history of their article (what does this mean?). If published, this will include your full peer review and any attached files.

Reviewer #1: No

Reviewer #2: No

---

## [Author Response · Author response to Decision Letter 0]

25 Mar 2023

Dear Editor, 

Thank you for the opportunity to revise our manuscript. We appreciate the reviewers’ time and suggestions. Below please find a point-by-point response to their comments. 

Reviewer 1: 

Minor issues:

- A subgroup of postmenopausal women showed a relationship between UC and the vaginal microbiome in regard to a significant difference in alpha diversity and lack of Lactobacillus dominance. These results are not mentioned in the abstract, which might hide a relevant difference in a UC subgroup. We however agree with the central statement that menopausal status had a larger impact on vaginal microbial communities than IBD status.

We have now added the following statement to the abstract, lines 67-69: “A subgroup of postmenopausal women with Ulcerative colitis showed a significant higher alpha diversity and a lack of Lactobacillus dominance in the vaginal microbiome.”

- Line 177: Instead of a Poisson model for microbiome studies often a negative binominal model is used because microbiome data are overdispersed. Also, the parameters of the model are not given. It would be good to explain the choice of the model and provide details of the model formula. Please state which parameters were used in the model formula (timepoint? IBD status? Patient ID?)

We thank the reviewer for this suggestion. Because our outcome is a categorical one, we repeated the analysis with a mixed effect logistic regression model including IBD status, with participant ID as the clustering variable. This has been described in the methods section on lines 194-208: “Using all samples from premenopausal participants, overall Lactobacillus dominance (as a categorical variable) was compared across IBD status using a mixed effects logistic regression, with participant ID as a grouping variable to account for multiple samples from some participants.” The model results are now reported as odds ratios in Table 2

 Line 151: Please provide information given about which bacteria database was used for taxonomy matching to the ASVs before the BLAST step (Greengenes, SILVA, custom?) 

We now include the following in the methods section on lines 154-156: “The Assign Taxonomy function was used to implement RDP Naive Bayesian Classifier algorithm as previously described27 with kmer size 8 and 100 bootstrap replicates.”

 Line 211: Because vaginal swabs are less diverse than fecal samples, samples cluster mainly in regard to the major Lactobacillus subtype/ non-Lactobacillus status. This could also be mentioned in the text to clarify for the readers.

We now state the following on lines 184-187: “Since the diversity of the vaginal microbiome is limited and Lactobacillus species make up over 90% of the microbial composition, vaginal microbial communities generally cluster according to the dominant Lactobacillus species. Some studies cluster samples by Community State Type (CST),9 and others use a Lactobacillus relative abundance threshold to classify communities.11,13”

 Line 288: The discussion lacks any consideration of the health of the vaginal microbiome. In contrast to the gut microbiota where a higher diversity is generally regarded as beneficial, this might not be the case for the vaginal microbiome, as a diverse microbiota could be an indication of bacterial vaginosis. Thus, an increased alpha-diversity in a subgroup (potentially by cross-contamination) is a hypothetical pathogenetic mechanism in accordance with the presented results in postmenopausal UC patients. We propose to discuss this also in the text.

We have now added the following sentence on lines 362-363: “In contrast to the gut, where a diverse microbiome is considered beneficial, a healthy vaginal microbiome has low diversity and is dominated by lactobacilli.”

Additionally, on lines 365-368 we state: “The exception to this was in the subset of postmenopausal women with Ulcerative colitis. However, with such small numbers it is challenging to ascertain whether the difference in microbiota was due to menopausal status or underlying IBD.”

Reviewer #2: 

Methods

L151: The authors should indicate which classifier and database were used for ASV assignment as well as parameters set in the DADA2 pipeline applied

We now include the following in the methods section on lines 154-156: “The Assign Taxonomy function was used to implement RDP Naive Bayesian Classifier algorithm as previously described27 with kmer size 8 and 100 bootstrap replicates.”

LL168-177. The authors should indicate which software was used for analysis of alpha- and beta-diversity. In general, this section is too short to be able to reproduce the analysis of raw data and thus, should either be specified with parameters used (filtering of low abundant samples, evaluation of controls, etc.) or provide codes used in an appropriate online repository, e.g. Github.

We now include the following: 

Line 162: “Taxa that were present in less than 5% of the samples were removed.”

Line 177-180: “The R software was used for statistical analysis, using Ape and Vegan packages. Alpha diversity was assessed using the Shannon Diversity Index and was calculated using Diversity() function which is part of the Vegan package. Beta diversity was calculated using the Vegdist() and Pcoa() functions.”

Results

LL211-213: The existence of those clusters has been described earlier and have been classified as Community state types (CSTs) (https://www.ncbi.nlm.nih.gov/pmc/articles/PMC3063603/). This information should be added and probands should be classified in those types.

We have now added the following on lines 186-187: “Some studies cluster samples by Community State Type (CST),9 and others use a Lactobacillus relative abundance threshold to classify communities.11,13 We assigned a categorical variable based on relative abundance of Lactobacillus species...”

Discussion

LL321-323: Since this finding is based on a very low number of samples, the authors should avoid to give any therapy indications but rather stick to the follow-up studies with more individuals included as suggested in the following section.

We have now taken out the clinical recommendation so the sentence on lines 383-384 says: “"The low prevalence of Lactobacillus seen in postmenopausal women with IBD could put them at higher risk for recurrent UTI.”

LL329-333: This is very speculative – could the authors include some literature underlining this hypothesis? Particularly, as the authors could not detect any significant changes between active IBD and remission, this seems to be overstated.

We agree with the reviewer that this is speculative. Our argument here is not that there is a difference in microbiota in the vagina, but that there may be a difference in the immune response to microbes that is universal across mucosal surfaces. We have tried to make this more clear by starting the sentence on line 387 with “We hypothesize that a difference might be found in the vaginal mucosal immune function rather than in the vaginal microbiome and so a potential next step could be to…”

Additionally, we start the sentence on line 391 with: “If indeed there is a difference in the mucosal immune function then…” 

LL335-345: pH is the most common factor that at least drive differences between Lacto and non-Lacto groups. Thus, it should be included within the next study.

We agree with the reviewer that pH is a characteristic difference between these communities, however, it is as a result of the presence or absence of lactobacilli rather than a driver of those differences.(See O’Hanlon PMID 30642259)

Thank you again for the opportunity to improve our manuscript.

Sincerely,

Ofri Bar

Caroline Mitchell

---

## [Decision Letter · Decision Letter 1]

6 Apr 2023

Comparison of vaginal microbiota between women with inflammatory bowel disease and healthy controls

PONE-D-23-01900R1

Dear Dr. Mitchell,

We’re pleased to inform you that your manuscript has been judged scientifically suitable for publication and will be formally accepted for publication once it meets all outstanding technical requirements.

Kind regards,

José António Baptista Machado Soares, PhD

Academic Editor

PLOS ONE

Additional Editor Comments (optional):

Dear authors,

I am glad to say that both reviewers acepted the revised version of the present study. Therefore, I also endorse the publication of the present version. Thank you and congratulations of an excellent work.

Best regards,

António

Reviewers' comments:

Reviewer's Responses to Questions

**Comments to the Author**

1. If the authors have adequately addressed your comments raised in a previous round of review and you feel that this manuscript is now acceptable for publication, you may indicate that here to bypass the “Comments to the Author” section, enter your conflict of interest statement in the “Confidential to Editor” section, and submit your "Accept" recommendation.

Reviewer #1: All comments have been addressed

Reviewer #2: All comments have been addressed

2. Is the manuscript technically sound, and do the data support the conclusions?

Reviewer #1: (No Response)

Reviewer #2: Yes

3. Has the statistical analysis been performed appropriately and rigorously? 

Reviewer #1: (No Response)

Reviewer #2: Yes

4. Have the authors made all data underlying the findings in their manuscript fully available?

Reviewer #1: (No Response)

Reviewer #2: Yes

5. Is the manuscript presented in an intelligible fashion and written in standard English?

Reviewer #1: (No Response)

Reviewer #2: Yes

6. Review Comments to the Author

Reviewer #1: (No Response)

Reviewer #2: (No Response)

7. PLOS authors have the option to publish the peer review history of their article (what does this mean?). If published, this will include your full peer review and any attached files.

Reviewer #1: No

Reviewer #2: No

---

## [Editor Report · Acceptance letter]

24 Apr 2023

PONE-D-23-01900R1 

Comparison of vaginal microbiota between women with inflammatory bowel disease and healthy controls 

Dear Dr. Mitchell:

I'm pleased to inform you that your manuscript has been deemed suitable for publication in PLOS ONE. Congratulations! Your manuscript is now with our production department. 

Kind regards, 

on behalf of

Dr. José António Baptista Machado Soares 

Academic Editor

PLOS ONE